# FEATURE–LABEL EMBEDDING ALIGNMENT FOR BACKPROP-FREE TINY NETWORKS

## ABSTRACT

Forward-only training methods, such as Forward-Forward (FF) learning, provide a lightweight alternative to backpropagation by eliminating the backward pass. This reduces memory usage and makes them well-suited for forward-propagation accelerators, which are increasingly common in modern embedded devices. However, current forward-only methods struggle to scale to deep networks and challenging visual recognition tasks, resulting in a substantial performance gap compared to backpropagation. We address this limitation with the *Feature-Label Embedding Alignment (FLEA)* block, a novel architectural component that allows FF networks to scale effectively while remaining forward-only. FLEA introduces *layer-wise discriminative learning*, where each layer independently optimizes its parameters by aligning its feature embedding with the corresponding label embedding that maximizes class separability. This produces more discriminative representations in deeper layers and significantly narrows the performance gap with backpropagation. Experiments demonstrate that FLEA-equipped FF networks achieve competitive accuracy on complex visual benchmarks while retaining the memory efficiency and accelerator compatibility that make forward-only training appealing for resource-constrained systems.

## 1 INTRODUCTION

Training deep neural networks on resource-constrained embedded devices remains a fundamental challenge in computer vision. The standard backpropagation algorithm, although effective at achieving state-of-the-art performance, presents critical incompatibilities with embedded hardware because of its substantial memory overhead and computational complexity. Backpropagation requires storing intermediate activations during the forward pass and performing gradient computations during the backward pass, typically demanding memory that scales linearly with network

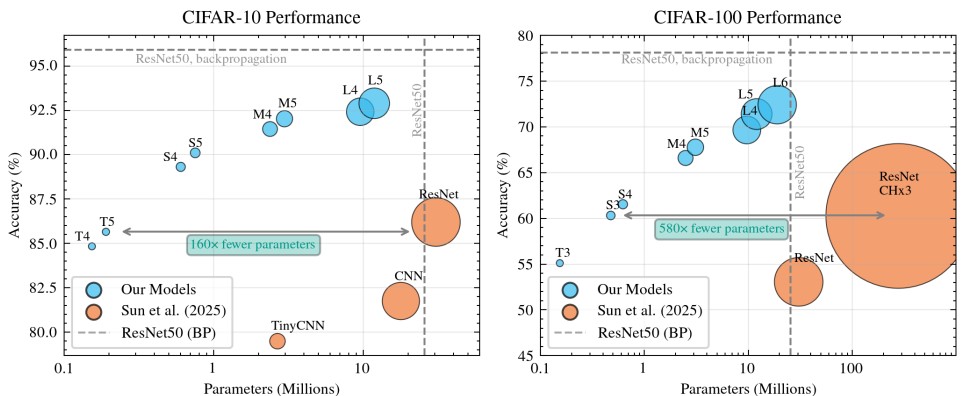

Figure 1: Performance of the proposed forward-only implementation compared to Sun et al. (2025). Models are detailed in Table 1 and Table 2

.

depth. This often exceeds the hundreds of kilobytes of SRAM available on typical microcontroller units (MCUs) by orders of magnitude. Furthermore, many recent embedded devices include specialized convolutional accelerators optimized for inference but lacking support for some operations required by backpropagation, such as transposed convolutions and high-precision gradient accumulation. Existing approaches for on-device learning primarily focus on reducing backpropagation's computational and memory demands via gradient compression, activation checkpointing (Vogels et al., 2019), or partial parameter updates (Paissan et al., 2024). However, these methods still depend on a backward pass and may exceed the capabilities of ultra-low-power systems—particularly for deeper networks or complex visual recognition tasks. Forward-only training methods represent a paradigm shift by eliminating the backward pass, significantly reducing memory footprint and computational complexity while preserving compatibility with inference-optimized hardware. Approaches such as Forward–Forward (FF) learning (Hinton, 2022) and error-driven methods like PEPITA (Dellaferrera & Kreiman, 2022; Srinivasan et al., 2024) have shown competitive performance on small-scale image classification tasks with shallow networks. However, these methods often degrade when scaled to more challenging tasks requiring deeper architectures and larger output spaces, limiting their practical deployment. This scalability challenge stems from the difficulty of enabling effective layer-wise learning without global gradient information, which can lead to suboptimal feature representations in deeper layers. In addition, current FF approaches can incur substantial inference overhead, requiring either large computational cost (Hinton, 2022) or increased parameter count (Sun et al., 2025).

To address these limitations, we propose *FLEA* (Feature–Label Embedding Alignment), a novel architectural block that enables forward-only networks to scale to complex recognition tasks while preserving their advantages for embedded deployment. Our approach introduces *layer-wise discriminative learning*, where each block independently optimizes features to maximize class separability by aligning learned representations of inputs and labels. This maintains the memory efficiency and hardware compatibility of forward-only training while enabling continuous network updates without the update locking required in backpropagation. Crucially, as we demonstrate empirically in Section 5, our method scales without increasing parameter count (Sun et al., 2025) or inference complexity (Hinton, 2022), addressing key limitations of existing forward-only approaches.

Our main contributions include: (1) a theoretical framework explaining Forward–Forward learning dynamics; (2) the FLEA block architecture derived from this framework; (3) an efficient one-shot inference technique without additional parameter count or computational overhead relative to comparable baselines; and (4) empirical evidence that FLEA achieves performance competitive with backpropagation on more challenging benchmarks than the current state of the art.

We begin by reviewing related work and explaining the principles of forward-forward learning in Sections 2 and 3. Our proposed methodology is described in Section 4, and Section 5 presents an experimental evaluation comparing the performance of our approach with existing state-of-the-art.

## 2 PREVIOUS WORKS

Training neural networks directly on severely memory-constrained devices (e.g., microcontrollers with 100–500 KB SRAM) is challenging because of the high memory, latency, and energy cost of backpropagation. The backward pass requires storing intermediate activations and gradients, often increasing memory usage by over $6\times$ compared to inference (Lin et al., 2023; Hinton, 2022; Chakrabarti & Moseley, 2019), which is incompatible with real-time, power-limited edge deployments (Llisterri Giménez et al., 2022).

To address these constraints, several backpropagation-free or locally updated training methods have been explored. Forward–Forward (FF) (Hinton, 2022) replaces the backward pass with two forward passes: a *positive* pass using real data and a *negative* pass using corrupted inputs. Each layer independently optimizes a *goodness* function, enabling learning without storing intermediate activations. FF achieves competitive accuracy on classification tasks with much lower memory requirements and has been adapted to convolutional networks (Deng et al., 2023; Aktemur et al., 2024; Scodellaro et al., 2023). Building on this, DeeperForward (Sun et al., 2025) extends FF to CNNs as deep as 17 layers by using layer normalization and mean-based goodness; this mitigates feature-scaling and neuron-deactivation issues and improves accuracy on MNIST, Fashion-MNIST, and CIFAR-10. The authors also propose model-parallel training that exploits FF's local updates for efficient distributed

learning. PEPITA (Dellaferrera & Kreiman, 2022; Srinivasan et al., 2024) (Present Error to Perturb Input To modulate Activity) offers another fully forward alternative. It injects output errors into the inputs through a fixed random feedback matrix and uses two forward passes to compute weight updates from activation differences, yielding a local Hebbian-like update. PEPITA attains competitive accuracy with minimal memory requirements and avoids weight transport. Other approaches such as Feedback Alignment (Lillicrap et al., 2014), Direct Feedback Alignment (Nokland, 2016), and Difference Target Propagation (Lee et al., 2015) relax backpropagation's weight-symmetry constraints but still rely on backward-phase computations and intermediate state storage. Equilibrium Propagation (Ernoult et al., 2020) improves biological plausibility but typically depends on iterative solvers or symmetric feedback pathways, limiting applicability to ultra-low-power devices. Overall, FF-based methods—especially recent advances like DeeperForward—and PEPITA stand out as promising solutions for continual and streaming learning on constrained edge devices due to their purely forward-pass operation, low memory footprint, and parallelization potential.

# 3 LEARNING REPRESENTATIONS AND FORWARD–FORWARD

Deep neural networks consist of multiple layers; the state of each layer represents the input in a *feature space*. Each layer transforms the representation received from the previous layer into a new representation and forwards it. In models trained with backpropagation, the last hidden layer's representation is used to perform the target task (for example, predicting class membership probabilities). The output layer's state is compared to the desired target and an error is computed via a loss function. A gradient-based optimizer then adjusts the parameters of the final layer to reduce this loss; by the chain rule, this *global error signal* propagates backward through the network and updates preceding layers until the first hidden layer is reached.

In classification, the classifier head typically contains one neuron per class. Each neuron evaluates how strongly the high-level features of the last hidden layer correlate with a given class, and the class whose neuron shows the strongest evidence is selected. As a consequence of training, internal "hidden" units (which are neither input nor output units) come to represent task-relevant features, and the network captures regularities in the task through interactions among these units. (Rumelhart et al., 1986; Goodfellow et al., 2016)

The *Forward–Forward* (FF) algorithm replaces the backward pass with a second forward pass. It constructs two types of inputs—*positive* and *negative*—and assigns higher *goodness* (or lower energy) to positive configurations and lower *goodness* (or higher energy) to negative configurations. We provide an analytical explanation that shows how the energy-based objective drives the first layer to extract useful features.

## 3.1 SUPERVISED FORWARD–FORWARD

We consider the supervised FF setting for a feed-forward network in which labels are embedded in the input, following Hinton (2022). Labels are encoded as one-hot vectors and concatenated with the input.[1] If the provided label matches the true label of the sample, the constructed input is *positive*; otherwise it is *negative*. For each hidden layer we define the *goodness* $\mathcal{G}$ as the sum of squared activations in that layer $\mathcal{G} = \sum H^2$, where $H = \max(0, Z)$ denotes the ReLU activations of the layer. The algorithm runs two forward passes per example: a *positive pass* (positive input) and a *negative pass* (negative input). An input is classified as positive if its goodness exceeds a threshold and as negative otherwise. Parameters are optimized with a gradient-based optimizer using a binary cross-entropy loss on the probability that the input is positive, where the probability is computed by applying a logistic function to $\mathcal{G} - \theta$.[2]

Because margin ranking loss provides results comparable to binary cross-entropy while allowing for a simpler explanation, we base the subsequent analysis on the margin loss. In this setting, the objective is that for each layer the goodness for the positive input, $\mathcal{G}^+$, exceeds the goodness for the negative input, $\mathcal{G}^-$, by at least a margin $m$. Formally, for a certain layer the margin ranking loss is

$$\mathcal{L} = \max\big(0,\; m - (\mathcal{G}^+ - \mathcal{G}^-)\big). \tag{1}$$

---

[1] Hinton (2022) uses the first 10 pixels of each MNIST sample to encode the label. For permutation-invariant models this is equivalent to concatenating the label vector to the input.

[2] Hinton (2022) uses the layer dimension $\theta = D$ as a default threshold.

The positive input is the pair of the sample with its true class (denoted as $\ell^+$ or $y$). For example $(x^y, y)$, i.e., sample $x$ with true label $y$. The negative input is the pair of the sample with a false class (denoted as $\ell^-$ or $\tilde{y}$). For example $(x^y, \tilde{y})$, i.e., sample $x$ belongs to class $y$ while $\tilde{y} \neq y$. False label $\tilde{y} \sim \mathcal{U}(\mathcal{Y} \setminus \{y\})$ is sampled uniformly from the subset of wrong classes.

## 3.2 FIRST-LAYER DYNAMICS AND INTERPRETATION

To understand how local updates using FF can lead to neurons learning specialized features even in the absence of a global error signal, we analyze the learning dynamics of the first hidden layer. We denote the pre-activation of neuron $i$ in the positive and negative passes as $z_i^{\pm}$, defined by

$$z_i^{\pm} = F_i(x) + L_i(\ell^{\pm}), \tag{2}$$

where $F_i(\cdot)$ is the *feature detector* and $L_i(\cdot)$ is the *label-conditioned bias*.

$$F_i(x) = W_{i,:}\, x + b_i, \quad L_i(\ell) = B_{i,\ell} \tag{3}$$

Here $W$ is the input weight matrix, $b$ is the global bias vector, $B$ is a label-conditioned bias matrix,[3] and $\ell$ is the provided label. Note that these two terms are independent: $F_i(x)$ does not depend on the label $\ell$ and $L_i(\ell)$ does not depend on the sample $x$. During the positive and negative passes the sample $x$ is the same and the only changing factor is the conditioned bias. Finally, we denote the activation of neuron $i$ by $h_i$ and define it as

$$h_i^{\pm} = \max(0, z_i^{\pm}).$$

Gradients are computed only for active neurons ($\mathbf{1}_{z_i > 0}$). When the hinge is active (i.e., the margin is violated), the derivatives of the margin loss (Eq. 1) with respect to the goodness terms are $\partial\mathcal{L}/\partial\mathcal{G}^+ = -1$ and $\partial\mathcal{L}/\partial\mathcal{G}^- = +1$. Since $\mathcal{G} = \sum_i h_i^2$ and $\partial\mathcal{G}/\partial h_i = 2\,h_i$, the parameter gradients (only when the hinge is active) become[4]

$$\frac{\partial\mathcal{L}}{\partial W_{i,j}} = -2\,h_i^+\,\mathbf{1}_{z_i^+ > 0}\,x_j \;+\; 2\,h_i^-\,\mathbf{1}_{z_i^- > 0}\,x_j, \tag{4a}$$

$$\frac{\partial\mathcal{L}}{\partial b_i} = -2\,h_i^+\,\mathbf{1}_{z_i^+ > 0} \;+\; 2\,h_i^-\,\mathbf{1}_{z_i^- > 0}, \tag{4b}$$

$$\frac{\partial\mathcal{L}}{\partial B_{i,y}} = -2\,h_i^+\,\mathbf{1}_{z_i^+ > 0} \quad \text{(for the true label } y\text{)}, \tag{4c}$$

$$\frac{\partial\mathcal{L}}{\partial B_{i,\tilde{y}}} = +2\,h_i^-\,\mathbf{1}_{z_i^- > 0} \quad \text{(for the false label } \tilde{y}\text{)}. \tag{4d}$$

If neuron $i$ is inactive, no update occurs. If it is active in the positive pass, the true-label bias $B_{i,y}$ and $b_i$ increase, and the weights will align themselves with the input sample, raising $z_i^+$. If it is active in the negative pass, the false-label bias $B_{i,\tilde{y}}$ and $b_i$ decrease, and the weights will anti-align themselves with the input sample, lowering $z_i^-$.

Analyzing training steps for the first hidden layer, we can categorize neurons based on their activation pattern for a given sample $x^y$ and a false label $\tilde{y}$:

**Active only in the positive pass:** This indicates that the true-label bias $B_{i,y}$ is higher than the false-label bias $B_{i,\tilde{y}}$, signaling that the neuron's feature is relevant for class $y$. The update will align the weights with the input sample and increase $B_{i,y}$ while leaving $B_{i,\tilde{y}}$ unchanged (since the neuron was inactive in the negative pass and produced no gradient for $B_{i,\tilde{y}}$).

$$\Delta W_{i,j} \propto x_j\, h_i^+, \tag{5a}$$

$$\Delta B_{i,y} \propto h_i^+ > 0. \tag{5b}$$

---

[3]We distinguish $B$ from $W$ notation for clarity. In practice, it is just a subset of $W$ columns.

[4]Updates scale with the activation magnitude $h_i$, and unlike BCE, the margin objective imposes a strict hard-margin update.

**Active only in the negative pass:** This indicates that the false-label bias $B_{i,\tilde{y}}$ is higher than the true-label bias $B_{i,y}$, signaling that the neuron's feature is irrelevant for class $y$. The network will correct this by anti-aligning $W_{i,:}$ with the input sample $x^y$ and decreasing $B_{i,\tilde{y}}$ while leaving $B_{i,y}$ unchanged (since the neuron was inactive in the positive pass and produced no gradient for $B_{i,y}$).

$$\Delta W_{i,j} \propto - x_j\, h_i^-, \tag{6a}$$

$$\Delta B_{i,\tilde{y}} \propto - h_i^- < 0. \tag{6b}$$

**Active in both positive and negative passes:** If both indicators in Equation 4a are active, then we have Equation 7a. Since, according to Equations 2, the only difference between the pre-activations in the positive and negative pass is the conditional biases, we reach Equation 7b. Thus, the net effect on the update of the feature-detection part is a function of the difference in conditional biases. If the true label's bias $B_{i,y}$ is larger than the false label's bias $B_{i,\tilde{y}}$, then the feature detector will align itself with the feature in the sample; otherwise it will anti-align. Also, since Equations 5b and 6b both hold, the true label's conditional bias will increase and the false label's conditional bias will decrease.

$$\Delta W_{i,j} \propto x_j\, (h_i^+ - h_i^-), \tag{7a}$$

$$\Delta W_{i,j} \propto x_j\, (B_{i,y} - B_{i,\tilde{y}}). \tag{7b}$$

This dynamic ensures that, according to Equation 7b, the weights of each neuron align primarily with samples belonging to those classes whose conditional biases exceed a random conditional bias; the alignment strength is proportional to the difference between the true and false conditional biases. On the other hand, according to Equations 6b and 5b, the conditional biases correlate with the presence and magnitude of the features for their respective classes.[5]

This synergy creates an efficient division of labor: the $B$ matrix provides a "top-down" prior based on the class, while the $W$ matrix performs "bottom-up" feature extraction from the data. The training dynamic ensures they continuously co-adapt, with the positive pass reinforcing correct associations and the negative pass pruning away spurious ones. This is not merely a bias shift but the emergence of a structured internal representation where information about inputs and labels is seamlessly integrated. Overall, this contrastive process induces a powerful organizational principle within the network: neurons evolve from undifferentiated feature detectors into *specialized units*. The positive pass produces sparse activations since the feature detector $F(x)$ and class prior $L(y)$ are positively correlated. In the negative pass, $F(x)$ and $L(\tilde{y})$ are typically uncorrelated or negatively correlated, yielding weaker responses.

During inference, we try all labels' conditional biases and choose the label that yields the highest *goodness*. Because the activation function is applied after the addition and only active neurons contribute, we cannot directly measure alignment between $F(x)$ and $L(\ell)$.

### 3.3 ON THE INSEPARABILITY OF THE POSITIVE AND NEGATIVE PHASES

Hinton (2022) observes that "using the sum of squared activities as the goodness function, alternating between thousands of weight updates on positive data and thousands on negative data only works if the learning rate is very low and the momentum is extremely high," calling this "probably the most important outstanding question about FF as a biological model." We do not attempt to explain this empirical requirement here. Our analysis below focuses on why interleaved positive/negative updates promote feature selectivity locally, but it does not resolve why thousands of segregated updates require very low learning rates and high momentum in practice. We therefore treat Hinton's observation as an important empirical constraint and leave a systematic investigation of optimization dynamics under phase separation to future work.

With many positive-only updates, as Equation 5b shows the conditional bias matrix $B$ grows for all labels, driving neurons to activate indiscriminately. Also, as Equation 5a shows, the feature-detection matrix $W$ aligns itself with all input samples indiscriminately, thus losing sparsity and

---

[5]The conditional bias matrix $B$ learns a statistical summary of the dataset, where $B_{i,y}$ comes to represent the correlation between the feature captured by neuron $i$ and class $y$, i.e., how much the detected feature is present in the samples belonging to class $y$ rather than the other classes. A high $B_{i,y}$ means the feature is a strong predictor of that class. The details and the empirical evidence are provided in Appendix D.

specialization. Subsequent negative-only updates, according to Equation 6b, simply shrink $B$ and, according to Equation 6a, cause $W$ to anti-align with all input samples without correcting spurious correlations, since neurons are no longer feature-selective. Interleaving positive and negative passes avoids this collapse: positive updates strengthen class-relevant features, while negative updates prune misaligned associations in real time.

## 3.4 CONCISE INTUITION

Forward–Forward produces specialization through cooperation between bottom-up detectors $W$ and top-down class priors $B$. The positive pass strengthens feature–class associations and aligns $W$ with class-relevant inputs; the negative pass weakens associations that do not co-occur with the true label. The result is a structured internal representation in which a small subset of neurons exhibits sparse, high-amplitude responses for the true class.

The Forward–Forward contrastive objective thus drives the co-evolution of top-down class priors and bottom-up detectors, yielding sparse, class-specific activations that support reliable classification without backpropagation.

## 4 METHODOLOGY

Based on the observations in Section 3.2, we propose that instead of adding the signal coming from the label (i.e., treating $B$ as a conditional bias) to the feature representation (i.e., $F(x)$), we can treat $B$ directly as a label encoder that embeds the label into the feature space. During training we then use the alignment between the true/false label embedding and the representation embedding of the input sample as the contrastive objective. Because the supervisory signal tends to vanish with depth, we apply this alignment at every layer. In this *layer-wise discriminative learning* regime, each layer independently learns representations that maximize class separability. To make per-layer supervision feasible, each layer compresses its feature maps into a compact embedding (either globally or over fixed spatial partitions). Despite the absence of a global error signal, intermediate layers still produce representations that are sufficiently informative for deeper layers.

Because the alignment occurs after the activation function and supervisory signals do not flow through the network, inference can be done in a single shot: there is no need to try all possible labels (which would scale with the number of labels) nor to rely on a separate classifier head that requires the full forward pass through the entire network. This simplification follows directly from the linearity of the label encoder, which allows the score for all classes to be computed in a single matrix–vector product.

### 4.1 MODEL OVERVIEW

The network stacks multiple `FLEA` blocks (optionally with pooling and skip connections). At inference each block behaves like a standard convolutional layer. During training each block provides a per-layer discriminative objective by (i) encoding its feature maps into a compact *representation embedding* and (ii) embedding class labels into the same space via a *label encoder*. Training aligns the feature embedding with the true-label embedding and pushes it away from selected false-label embeddings; at inference, the predicted label is the one whose embedding has maximum alignment with the layer's representation embedding.

### 4.2 FLEA

Each block contains a standard convolutional *feature extractor* followed by normalization and non-linearity, together with a *representation encoder* and a *label encoder* used for per-layer supervision.

**Feature extractor.** Given input activations, we first apply LayerNorm, then a convolution, followed by BatchNorm (optionally without affine parameters) and a ReLU non-linearity. The resulting feature maps $z \in \mathbb{R}^{C_{\text{out}} \times H \times W}$ are passed forward and also reduced to produce a compact embedding for supervision.

**Representation encoder.**   We aggregate each channel's activations to produce a vector that quantifies feature presence. Let $\{\Omega_p\}_{p=1}^{P}$ be an optional partition of the spatial grid (with $P = 1$ for global pooling). The representation $\phi(z) \in \mathbb{R}^D$ (where $D = C_{\text{out}} \times P$) is

$$\phi(z) = \left[ \frac{1}{|\Omega_q|} \sum_{(h,w) \in \Omega_q} z_{c,h,w} \right]_{\substack{c=1,\ldots,C_{\text{out}} \\ q=1,\ldots,P}} \in \mathbb{R}^D.$$

**Label encoder.**   Each layer has a linear label encoder that maps a one-hot class vector $\mathbf{e}^{(\ell)} \in \mathbb{R}^N$ into the representation space:

$$c(\ell) = B\mathbf{e}^{(\ell)} + b = B_{:,\ell} + b \in \mathbb{R}^D, \qquad B \in \mathbb{R}^{D \times N}, \; b \in \mathbb{R}^D,$$

where $B$ is the label-encoder weight matrix.  Intuitively, $c_i(\ell)$ assigns per-dimension relevance scores for class $\ell$ at layer $i$.  The bias $b$ lets the layer adjust the overall goodness scale without changing relative alignments.

**Goodness.**   Goodness $\mathcal{G}$ is defined as the dot product between the representation embedding $\phi(x)$ and the label embedding $c(\ell)$.  It measures how aligned the *representation embedding* and *label embedding* are.

**Stochastic Negative Label Sampling.**   Hinton (2022) propose selecting a false label uniformly at random among all incorrect classes to construct negative samples.  Inspired by Robinson et al. (2021), we instead adopt a *hard negative mining* approach so the model focuses on more informative and confusing alternatives.

We define a *hard* negative label as one that receives a high score from the current layer despite being false.  Rather than selecting this label deterministically (e.g., via 'argmax'), we introduce stochasticity by sampling the false label from a soft distribution over all false classes, weighted by their predicted scores.  The detail of the hard negative mining are given at Appendix E.

### 4.3   TRAINING OBJECTIVE

Training consists of a positive and a negative pass. In the positive pass, at layer $i$, for an input–label pair $(x, y)$ we compute the representation embedding $\phi_i(x)$, label embedding $c_i(y)$, and then the goodness $\mathcal{G}_i^{+} = c_i(y) \cdot \phi_i(x)$. In the negative pass, we compute $\mathcal{G}_i^{-} = c_i(\tilde{y}) \cdot \phi_i(x)$.

Per-layer binary cross-entropy loss can be written as

$$\mathcal{L} = -\log \sigma\big(\mathcal{G}^{+}\big) - \log\big(1 - \sigma(\mathcal{G}^{-})\big).$$

For simplicity of exposition, we write derivatives using the margin ranking loss from Equation 1 instead of binary cross-entropy. The only difference is that under BCE the confidence of the prediction modulates gradient magnitudes, while the update directions remain the same:

$$\frac{\partial \mathcal{L}}{\partial c_i(y)} = -\phi_i(x), \qquad \frac{\partial \mathcal{L}}{\partial c_i(\tilde{y}_i)} = \phi_i(x), \tag{8}$$

$$\frac{\partial \mathcal{L}}{\partial \phi_i(x)} = c_i(\tilde{y}_i) - c_i(y). \tag{9}$$

These expressions hold under the margin-loss view; with BCE the magnitudes are modulated by confidence but the directions remain identical.

Note that these derivatives apply when the margin is violated (i.e., when the layer fails to make the positive goodness exceed the negative goodness by at least the margin). In that case the updates move $c_i(y)$ toward $\phi_i(x)$ and $c_i(\tilde{y})$ away from $\phi_i(x)$; simultaneously $\phi_i(x)$ moves toward $c_i(y)$ and away from $c_i(\tilde{y})$. Effectively, these updates align $\phi_i(x)$ with $c_i(y)$ and anti-align it with $c_i(\tilde{y})$. This dynamic is closely analogous to the analysis in Section 3.2 (Equations 5b and 6b are analogous to Equation 8 and Equation 7b to Equation 9).

## 4.4 INFERENCE

At inference, the representation embedding $\phi(x) \in \mathbb{R}^D$ is projected into the label space to produce the class-score vector

$$s = B^\top \phi(x) \in \mathbb{R}^N,$$

whose $\ell$-th entry is $s_\ell = \phi(x)^\top B_{:,\ell}$. The predicted label is $\hat{y} = \arg\max_\ell s_\ell$. This computation is a single matrix–vector product and thus has cost comparable to a standard linear classifier. By contrast, the native Forward–Forward inference requires a separate forward pass per candidate label, which becomes impractical as the number of classes grows. Thus, our inference requires a single $\mathcal{O}(DN)$ matrix–vector product per input, rather than performing $N$ full forward passes through the network as in label-conditioned Forward–Forward (i.e., roughly $N$ times more expensive).

Since each layer produces its own prediction we can choose the final depth based on validation accuracy and model size, and discard deeper layers if they do not contribute improvements. In this case, only the label encoder corresponding to the chosen final layer will be retained.

## 4.5 KEY IMPLICATIONS

Our implementation of the Forward–Forward framework yields several practical advantages over standard backpropagation and its more biologically plausible variants:

**Elimination of activity freezing:** Unlike algorithms that rely on top-down error propagation, our method requires no activity freezing in the absence of a backward pass. This removes the need to store intermediate activations for a separate learning phase, eliminating a major memory bottleneck of backpropagation (Barley & Fröning, 2024).

**Inherently local and parallelized learning:** Each layer is optimized directly through its own goodness score, enabling two key benefits: (1) *local training*, where layers update based on local signals without waiting for downstream computations; and (2) *no update locking*, which allows parameter updates at one layer while data flows forward to the next. This independence of updates can significantly improve training efficiency.

**Class-adaptive feature specialization:** The layer-wise objective encourages neurons to develop class-adaptive features. This supports a hierarchical training strategy where coarse-grained superclasses train early layers. For instance, ImageNet's natural hierarchy (Deng et al., 2009)—comprising 10 superclasses each containing 7–116 related subclasses (e.g., 52 bird species, 116 dog breeds)—provides a convenient label grouping for initial blocks. These layers capture low-level features shared across semantically related categories within a superclass.

**Mitigation of vanishing signals:** A known limitation of the original Forward–Forward algorithm is the rapid attenuation of the supervisory signal with depth, as the distinction between positive and negative passes diminishes. Our approach injects label information at every block, ensuring a consistent, strong training signal throughout deep networks. By contrast, alternative deep Forward–Forward methods like DeeperForward (Sun et al., 2025) dedicate a separate group of filters to each class at every layer; this approach hinders shared feature learning, leading to parameter redundancy and reduced compactness.

## 5 EXPERIMENTS

We evaluate our model on CIFAR-10 and CIFAR-100 across four model sizes—Tiny, Small, Medium, and Large—each available in 3-, 4-, or 5-block variants. The model architectures and computational costs are detailed in Appendix A.

We use the standard CIFAR training split of 50K samples, with 45K for training and 5K for validation. The original 10K samples of the test set are used to report test accuracy. We fine-tuned hyperparameters on the validation set and report mean test accuracy and standard deviation over 10 independent runs (seeds 0–9). Comprehensive training details, including data-augmentation specifications and hyperparameters, are provided in Appendix B.

Table 1: CIFAR-10 test accuracy (%). Reported numbers for our models are averaged over 10 runs with different seeds; baseline results are taken from the respective papers.

| Model | Parameters | Test Accuracy (%) |
|---|---|---|
| Tiny (4-block) | 153K | $84.85 \pm 0.23$ |
| Tiny (5-block) | 190K | $85.66 \pm 0.27$ |
| Small (4-block) | 602K | $89.32 \pm 0.21$ |
| Small (5-block) | 749K | $90.12 \pm 0.21$ |
| Medium (4-block) | 2.38M | $91.46 \pm 0.15$ |
| Medium (5-block) | 2.97M | $92.05 \pm 0.21$ |
| Large (4-block) | 9.49M | $92.44 \pm 0.14$ |
| **Large (5-block)** | **11.84M** | **$92.92 \pm 0.13$** |
| FF-MLRNN (Hinton, 2022) | 3.41M | 59 |
| PEPITA-CNN (Dellaferrera & Kreiman, 2022) | 361K | $56.33 \pm 1.35$ |
| TinyCNN (Sun et al., 2025) | 2.67M | $79.49 \pm 0.29$ |
| CNN (Sun et al., 2025) | 17.77M | $81.76 \pm 0.30$ |
| ResNet (Sun et al., 2025) | 30.60M | $86.22 \pm 0.17$ |

Table 2: CIFAR-100 test accuracy (%). Reported numbers for our models are averaged over 10 runs with different seeds; baseline results are taken from the respective papers.

| Model | Parameters | Test Accuracy (%) |
|---|---|---|
| Tiny (3-block) | 153K | $55.13 \pm 0.18$ |
| Small (3-block) | 475K | $60.33 \pm 0.32$ |
| Small (4-block) | 622K | $61.55 \pm 0.33$ |
| Medium (4-block) | 2.48M | $66.64 \pm 0.39$ |
| Medium (5-block) | 3.07M | $67.79 \pm 0.35$ |
| Large (4-block) | 9.67M | $69.68 \pm 0.35$ |
| Large (5-block) | 12.03M | $71.40 \pm 0.20$ |
| **Large (6-block)** | **18.92M** | **$72.45 \pm 0.29$** |
| PEPITA-CNN (Dellaferrera & Kreiman, 2022) | 3.59M | $27.56 \pm 0.60$ |
| ResNet (Sun et al., 2025) | 30.60M | $53.09 \pm 0.79$ |
| ResNet-CHx3 (Sun et al., 2025) | 275.41M | $60.28 \pm 1.02$ |

The primary goal of this work is to demonstrate that convolutional networks can be effectively trained with the Forward–Forward algorithm. We performed only a limited hyperparameter search and anticipate that performance could improve with more extensive optimization.

## 6  CONCLUSION

In this work, we introduced the *Feature–Label Embedding Alignment (FLEA)* block, a novel architectural component designed to address the scalability limitations of forward-only training methods. We demonstrated how, by aligning each layer's representation embeddings with class-specific label embeddings, the proposed approach improves the separability of representations throughout network depth and thereby improves generalization.

Results show that FLEA consistently narrows the performance gap with backpropagation while maintaining the memory efficiency and accelerator compatibility that make forward-only training attractive for embedded systems. Notably, we observed stronger scalability on CIFAR-100: deeper networks equipped with FLEA retained discriminative capacity even as the number of classes increased. In our experiments, FLEA matched or exceeded the performance of existing state-of-the-art approaches with $580\times$ reduction in parameter count for CIFAR-100, while 2 networks with similar parameter count demonstrated a 20% accuracy increase when using FLEA.

Overall, FLEA represents a step toward practical, scalable forward-only training on resource-constrained hardware, reducing the need for backpropagation while achieving competitive performance on challenging visual benchmarks.

## REPRODUCIBILITY STATEMENT

All novel results presented in this paper are fully reproducible using the code provided as supplementary material. Our implementation includes scripts for model training and evaluation, with default configurations matching those reported in the paper. Hyperparameter settings, training schedules, and evaluation protocols are detailed in the main text and appendix. For comparison results, we report numbers directly from the respective original papers.

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

# A    ARCHITECTURE DETAILS

This appendix provides a comprehensive description of the model architecture. The network is built from a core `FLEA` block, which is applied sequentially with feature pooling and skip connections as described below.

## A.1    BLOCK SPECIFICATION

Each `FLEA` block contains a convolutional layer with a fixed number of filters $C$, which defines the model scale: Tiny ($C = 64$), Small ($C = 128$), Medium ($C = 256$), and Large ($C = 512$). All convolutions use $3 \times 3$ kernels with stride 1 and padding 1, preserving spatial dimensions.

## A.2    NETWORK DATA FLOW

Let $X_0$ be the input tensor and $X_i$ the output of the $i$-th block.

- **Block 1:** $X_1 = \text{FLEA}(X_0)$.
- **Block 2:** $X_2 = \text{FLEA}([X_0, X_1])$, where $[\cdot, \cdot]$ denotes concatenation along the channel dimension.
- **Block 3:** First concatenate the outputs of the first two blocks, $[X_1, X_2]$, then apply a $2 \times 2$ max-pooling (stride 2) to reduce spatial dimensions: $X_3 = \text{FLEA}(\text{MaxPool2d}([X_1, X_2]))$.
- **Block 4:** Apply average pooling to $X_3$ before processing: $X_4 = \text{FLEA}(\text{AvgPool2d}(X_3))$.
- **Block 5:** Apply average pooling to $X_4$ before processing: $X_5 = \text{FLEA}(\text{AvgPool2d}(X_4))$.

For the 6-block Large model we insert an extra block after Block 2. Concretely, the extra block processes $[X_0, X_1]$:

$$X_{\text{Extra}} = \text{FLEA}([X_0, X_1, X_2]).$$

Block 3 then consumes the concatenation of the first three block outputs, followed by max-pooling:

$$X_3 = \text{FLEA}(\text{MaxPool2d}([X_1, X_2, X_{\text{Extra}}])),$$

and Blocks 4 and 5 proceed as above.

## A.3    FLEA BLOCKS

Each `FLEA` block contains a convolutional feature detector whose output feature maps are passed to subsequent blocks. During training, two additional components are active: a *representation encoder* and a *label encoder*.

**Representation encoder.**    The representation encoder $\phi(\cdot)$ processes the feature maps produced by the convolutional layers. Each feature map is divided into $P$ non-overlapping regions (a hyperparameter). For each region and channel we compute the average of the activations, producing a representation embedding vector $\phi(z) \in \mathbb{R}^D$ with $D = C \times P$.

**Label encoder.**    The label encoder $c(\cdot)$ is a linear layer that projects a one-hot label vector $\ell$ into the same $D$-dimensional space, producing a label embedding $c_\ell \in \mathbb{R}^D$.

**Goodness.**    The goodness score for a feature map $z$ and a label $\ell$ is the dot product of their embeddings:

$$\mathcal{G}(\ell) = \phi(z) \cdot c(\ell). \tag{10}$$

During training, $\mathcal{G}$ is encouraged to be high for positive inputs (true label $\ell^+ = y$) and low for negative inputs (false label $\ell^- = \tilde{y}$). Figure 2 illustrates the active components of a `FLEA` block during training.

During inference, if the block is not producing a prediction, only the feature detector is active and forwards its feature maps to the next block. When the block performs active prediction, the representation encoder produces $\phi(z)$ and label scores are computed using the transposed label-encoder weights:

$$s = B^\top \phi(z) \in \mathbb{R}^N,$$

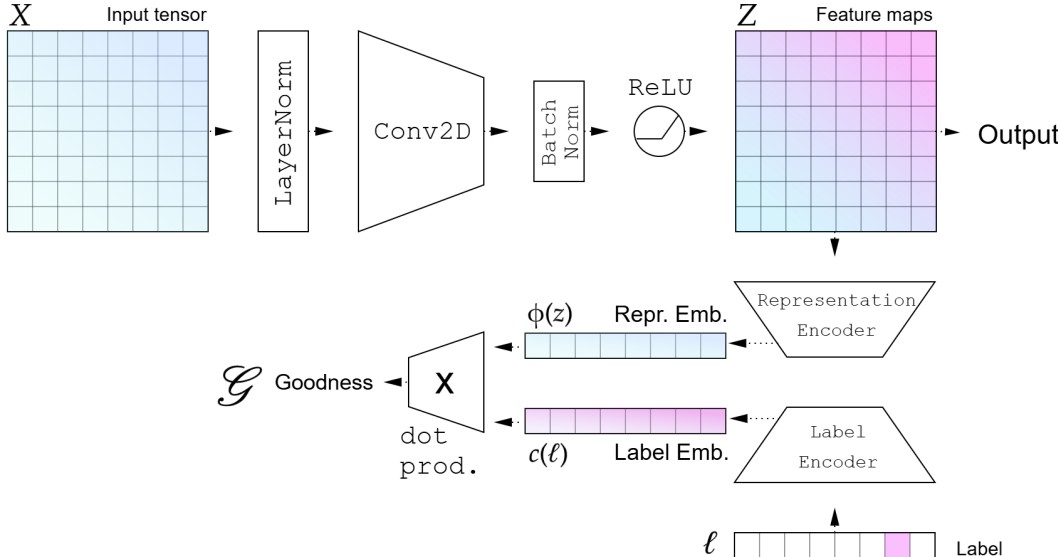

Figure 2: Training phase of a `FLEA` block. The representation encoder $\phi(\cdot)$ processes the feature maps to produce an embedding $z$, while the label encoder $c(\cdot)$ projects the label to an embedding $c_\ell$. The goodness $\mathcal{G}(\ell)$ is computed as their dot product.

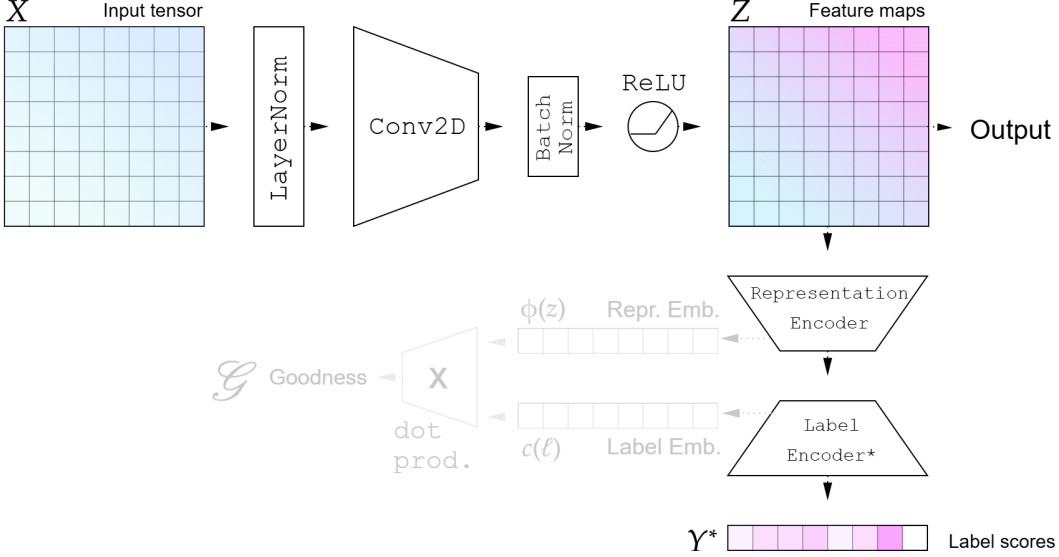

Figure 3: Inference phase of a `FLEA` block in active prediction mode. The representation embedding $\phi(z)$ is computed and multiplied by the transposed label encoder weights $B^\top$ to produce label scores.

where $B$ are the label-encoder weights. The label with the highest score is selected. Figure 3 shows the inference (prediction) mode. The computational cost of each model variant is reported in Table 3.

## B  HYPERPARAMETERS

We trained all models with AdamW and a polynomial decay learning-rate schedule (power = 0.9). The Conv2d backbone used an initial LR of $6 \times 10^{-3}$ for the first block and $1 \times 10^{-3}$ for the other blocks, with block-dependent weight decay ($10^{-3}, 10^{-2}, 10^{-1}$ for blocks 1–3, 4, and 5, respectively). The label encoder used initial LRs of $1.8 \times 10^{-3}, 3 \times 10^{-4}$, and $1 \times 10^{-4}$ (blocks 1, 2–4,

Table 3: Model computation cost for Tiny, Small, Medium, and Large variants. FLOPs are measured for a single forward pass on a CIFAR-10/CIFAR-100 sample.

| Model | Filters per Block | FLOPs |
|---|---|---|
| Tiny (4-block) | 64 | 63.05M |
| Tiny (5-block) | 64 | 63.64M |
| Small (4-block) | 128 | 244.04M |
| Small (5-block) | 128 | 246.41M |
| Medium (4-block) | 256 | 959.93M |
| Medium (5-block) | 256 | 969.39M |
| Large (4-block) | 512 | 3.81G |
| Large (5-block) | 512 | 3.85G |

Table 4: **Training hyperparameters.** Block-specific values are reported where applicable.

| Hyperparameter | Value |
|---|---|
| Optimizer | AdamW |
| Learning rate schedule | Polynomial decay (power = 0.9) |
| Conv2d initial LR | |
| first block | $6 \times 10^{-3}$ |
| other blocks | $1 \times 10^{-3}$ |
| Conv2d weight decay | |
| blocks 1–3 and Extra | $1 \times 10^{-3}$ |
| block 4 | $1 \times 10^{-2}$ |
| block 5 | $1 \times 10^{-1}$ |
| Label encoder initial LR | |
| first block | $1.8 \times 10^{-3}$ |
| blocks 2–4 and Extra | $3 \times 10^{-4}$ |
| block 5 | $1 \times 10^{-4}$ |
| Label encoder weight decay | $1 \times 10^{-3}$ |
| Partitioning | |
| blocks 1–3 and Extra | $4 \times 4$ |
| blocks 4–5 | $2 \times 2$ |
| Batch size | 128 |
| Epochs | 200 |
| RandAugment ops | 3 |
| RandAugment magnitude | 12 |
| Horizontal flip prob. | 0.5 |

and 5), with a uniform weight decay of $10^{-3}$. The negative-label hardness parameter $\alpha$ was set to 4, 6, and 8 for blocks 1, 2–3, and 4–5. Partitioning was $4 \times 4$ for the first three blocks and $2 \times 2$ for the last two. Training ran for 200 epochs with batch size 128, using RandAugment (3 ops, magnitude 12) and horizontal flips (probability 0.5).

## C  EXPERIMENTS ON MNIST

To verify that our method is effective even on simple datasets with compact architectures, we conducted experiments on MNIST (LeCun et al., 1998). All models were trained with the same hyperparameter configuration as in Table B, except for the 4-block model with 128 filters, which used no partitioning and fixed the negative label difficulty parameter $\alpha = 1$. For data augmentation, instead of the stronger transformations used in our main experiments, we applied only small rotations (up to 10 degrees) and slight shifts (up to 10%) in both directions to mimic natural variations in handwriting.

Table 5: MNIST test accuracy (%). Reported numbers for our models are averaged over 10 runs with different seeds; baseline results are taken from the respective papers.

| Model | Parameters | Test Accuracy (%) |
|---|---|---|
| 16 (2-block) | 5K | $99.10 \pm 0.05$ |
| 16 (3-block) | 10K | $99.45 \pm 0.07$ |
| 32 (5-block) | 48K | $99.59 \pm 0.05$ |
| 64 (5-block) | 188K | $99.65 \pm 0.05$ |
| **128 (4-block)** | **593K** | **$99.68 \pm 0.04$** |
| CNN (Sun et al., 2025) | 9.50M | $99.65 \pm 0.02$ |
| ResNet (Sun et al., 2025) | 26.83M | $99.63 \pm 0.04$ |
| TinyCNN (Sun et al., 2025) | 2.67M | $99.50 \pm 0.05$ |
| FF-FC (Hinton, 2022) | 13.61M | 99.36 |
| FF-CNN (Scodellaro et al., 2023) | 1.61M | 99.16 |
| FF-LRF (Hinton, 2022) | 3.03M | 98.84 |
| PEPITA-WM-FC (Srinivasan et al., 2024) | 821K | $98.42 \pm 0.05$ |
| PEPITA-CNN (Dellaferrera & Kreiman, 2022) | 260K | $98.29 \pm 0.13$ |

Table 5 reports the test accuracy averaged over 10 seeds. Our models achieve competitive or superior accuracy with orders of magnitude fewer parameters than baselines:

- With only 5K parameters (16 filters, 2 blocks), we reach 99.10%, which is close to or better than several much larger FF-based baselines.

- With 48K parameters (32 filters, 5 blocks), we achieve 99.59%, outperforming FF-FC (Hinton, 2022) by 0.23% while being about 280× smaller.

- With 188K parameters (64 filters, 5 blocks), we achieve 99.65%, matching the CNN baseline from Sun et al. (2025) while being nearly 50× smaller.

- Our best result, 99.68% with the 128-filter 4-block model (593K parameters), surpasses all other methods in Table 5. This model is nearly 16× smaller than the CNN baseline from Sun et al. (2025), despite achieving even higher accuracy.

These results show that our method scales down extremely well: even at tiny parameter budgets, it yields strong accuracy, and at moderate sizes it achieves state-of-the-art performance on MNIST.

## D ANALYZING THE CONDITIONAL BIASES

Equations 5b and 6b show that when a sample belongs to class $y$, the corresponding bias term $B_{i,y}$ receives positive updates proportional to the activation magnitude of neuron $i$. If false labels are sampled uniformly (balanced dataset), each label appears equally often as positive and as negative. An approximation for the expected update is therefore

$$\mathbb{E}[\Delta B_{i,y}] \; \propto \; \mathbb{E}[F_i(x) \mid y] \; - \; \frac{1}{|\mathcal{Y}| - 1} \sum_{k \neq y} \mathbb{E}[F_i(x) \mid k]. \tag{11}$$

Equation 11 should be understood as an approximation.

Thus $B_{i,y}$ increases when neuron $i$'s feature is, on average, more active for class $y$ than for other classes; otherwise it decreases. In this sense each bias term summarizes how strongly a neuron's feature correlates with a given class.

### D.1 EMPIRICAL VALIDATION

We validated this prediction on MNIST using a one-layer model with 1000 neurons.[6] After training we computed, for every neuron $i$ and class $y$, the conditional mean activation $\mathbb{E}[F_i(x) \mid y]$ on the

---

[6]Implementation based on Löwe's reimplementation of Forward–Forward (Hinton, 2022), available at https://github.com/loeweX/Forward-Forward. Hyperparameters and code were unchanged, except for appending one-hot labels after input normalization.

validation set and formed the contrast score given by the RHS of Equation 11. Correlating these contrast scores with the learned bias weights $B_{i,y}$ yields strong agreement: Pearson $r = 0.768$, Spearman $\rho = 0.705$ (two-sided $p < 10^{-6}$), supporting the hypothesis that conditional biases encode feature–class association strengths.

## D.2 NEURON ACTIVITY STATISTICS

We also analyzed per-neuron activation patterns at the final epoch (validation set averages):

- **Inactive in both passes:** $60.2\%$ of neurons (sparse representation).
- **Active in positive only:** $3.3\%$ of neurons; mean activation $\approx 0.36$.
- **Active in negative only:** $4.6\%$ of neurons; mean activation $\approx 0.36$.
- **Active in both passes:** $21.9\%$ of neurons; mean activations $\mathbb{E}[h_i^+] \approx 1.40$ and $\mathbb{E}[h_i^-] \approx 1.25$, gap $\approx 0.15$.

Within the group active in both passes, $58.4\%$ (i.e., $18.6\%$ of all neurons) satisfy $B_{i,y} > B_{i,\tilde{y}}$: these increase their mean activation from $1.20$ (negative) to $1.71$ (positive), an average rise of $+0.51$. The remaining $41.6\%$ of the both-pass group ( $13.3\%$ of all neurons) satisfy $B_{i,\tilde{y}} > B_{i,y}$ and decrease on average from $1.32$ to $0.97$ (change $-0.36$). This pattern suggests that while some features are shared across classes, features more predictive for the true class tend to have larger conditional biases and larger positive-pass activations.

## D.3 INTERPRETATION

The empirical results support three conclusions: (i) representations are sparse (majority of neurons inactive); (ii) a small fraction of neurons respond only in a single pass and with low amplitude; (iii) neurons active in both passes typically show stronger activation in the positive pass when the true-label bias dominates. Overall, conditional biases both reflect and reinforce feature–label associations as predicted by the theory.

# E HARD NEGATIVE MINING

We define a *hard* negative label as a wrong class that receives a relatively high score from the current layer. Instead of selecting this label deterministically (e.g., via `argmax`), we introduce stochasticity by sampling from a soft distribution over all incorrect classes.

To construct this distribution, scores of the classes are converted into log-probabilities using a softmax, then the true label is first excluded from consideration and renormalization yield the sampling distribution:

$$\tilde{p}_k = \frac{p_k}{\sum_{j \neq y} p_j}, \qquad k \neq y, \qquad \tilde{p}_y = 0,$$

where $p_k$ denotes the softmax probability of class $k$ before masking out the true label.

Finally, a false label $\tilde{y} \sim \text{Categorical}(\tilde{p}_1, \ldots, \tilde{p}_N)$ is drawn. This ensures that training consistently presents the model with diverse yet challenging negatives.

