# OpenReview forum: "Feature–Label Embedding Alignment for Backprop-Free tiny Networks"
_ICLR.cc/2026/Conference — Submitted to ICLR 2026_

### Official Review · Reviewer_LHKM · 2025-10-17

**Soundness:** 3
**Presentation:** 2
**Contribution:** 3
**Rating:** 4
**Confidence:** 3

**Summary:**

This paper enhances Forward-Forward (FF) learning — a forward-only alternative to backpropagation — by introducing the Feature-Label Embedding Alignment (FLEA) block. While FF methods reduce memory usage and suit hardware accelerators, they struggle to scale to deep networks and complex visual tasks. FLEA overcomes this limitation through layer-wise discriminative learning, aligning each layer’s feature embeddings with label embeddings to maximize class separability. This design enables FF networks to learn more discriminative representations, close the performance gap with backpropagation, and maintain the efficiency and hardware compatibility ideal for resource-constrained environments.

**Strengths:**

The paper proposed a novel method in Forward-Forward Learning, and provided systematic analysis from the preliminary background to their framework. Also they proved that their method is more efficient than conventional ways from the theoretical level. Based on their experiments, we could truly find their advantages than previous works and the experimental results are significant from the results they presented.

**Weaknesses:**

1) Firstly, even though they provided a detailed description of their method, the novelty is not enough. They are based on the basical chain rule and introduce another way on processing the 'label' in the traditional FF learning. The method is not so novel even if they gave a lot of descriptions. The main motivation aligns with the traditional FF learning principles and they did not design a novel method in FF learning. 2) Based on the experimental results, they truly present the significance of their method. However, they just performed experiments on CIFAR10/100, and also they just focused on the classification tasks. The models they used are convolutional NN. From my opinion, extensive experiments on other datasets, other tasks, and other various models not just CNN should be conducted to support their method's effectiveness.  Even if they offered the results on MNIST, it's a simple dataset and easier than CIFAR datasets. 3)  I am not sure if they compared to enough baselines in the relavant field. They just used 3 baselines in the main text. Considering the second weakness I proposed, I think the current experiments are not sufficient to support their method.

**Questions:**

N/A.

---

> ### Author Response · Authors · 2025-11-26
> **Response to Reviewer LHKM**
>
> We thank the reviewer for the detailed and constructive feedback. Below we address each point.
>
> ### **Response to weaknesses 1**
> We understand the reviewer’s concern regarding the perceived novelty of our approach; we would like to clarify that our contribution does not aim to modify Hinton’s Forward-Forward (FF) algorithm itself, but rather fundamentally rethink how it interacts with the model architecture. Our framework modifies the model blocks themselves so that FF learning can propagate meaningful supervisory signals throughout deep networks.
>
> As we hypothesize in lines 411–417, the original Forward-Forward (FF) algorithm suffers from rapidly diminishing supervisory signals as depth increases, which explains its limitation to very shallow networks (2–4 layers). This observation motivates our architectural design: instead of relying on a single global supervisory signal, we inject a local supervisory signal at every layer, enabling each layer to compute its own loss and update its parameters. This design choice fundamentally differs from the classical FF formulation and addresses its core scalability limitation.
>
> Furthermore, in backpropagation the gradients of each layer is calculated using the chain rule based on the gradient of the next layer. In our case, the loss function is between the feature extractor and the label encoder and no gradient is passes backwards between layers.
>
> ### **Response to weaknesses 2 and 3**
> We agree that extending FF-based learning to large-scale datasets (e.g., ImageNet) and tasks beyond image classification is an important direction for future research. A key challenge lies in the absence of inter-layer communication inherent to FF layer-wise training. The proposed label encoder provides a potential mechanism for enabling such communication. Furthermore, current literature on Forward-Forward is focusing on classification with MNIST and CIFAR-10, and we already show a significant push in that regard with CIFAR-100.

---

### Official Review · Reviewer_TDov · 2025-10-26

**Soundness:** 3
**Presentation:** 3
**Contribution:** 2
**Rating:** 4
**Confidence:** 3

**Summary:**

This paper proposes a novel architectural component, FLEA(Feature-Label Embedding Alignment), aimed at addressing the performance limitations of current forward-only methods, which struggle to scale to deep networks and challenging visual recognition tasks. FLEA introduces layer-wise discriminative learning, where each layer independently optimizes its parameters. The experiments on CIFAR-10/100 and MNIST show that FLEA-equipped FF networks achieve competitive accuracy on complex visual benchmarks.

**Strengths:**

- This paper introduces the FLEA block, a new architectural component designed to overcome the limitations of existing forward-only models, which often struggle to scale effectively to deeper networks and more complex visual recognition tasks. Building upon this module, the authors further develop a complete network architecture and a tailored training strategy. Together, these innovations enable the model to achieve substantial performance improvements across multiple benchmarks, clearly surpassing prior forward-only approaches.
- Beyond empirical results, the paper offers a theoretical analysis of the recognition capability of forward-only models, shedding light on their representational properties and limitations. This analytical perspective provides valuable guidance for future research, helping to establish a more principled foundation for advancing the understanding and development of forward-only neural architectures.

**Weaknesses:**

- The experimental evaluation primarily focuses on CIFAR-10 and CIFAR-100, with no tests on other datasets such as ImageNet, COCO, or more complex visual recognition tasks. This narrow scope raises questions about the generalizability of the proposed approach to diverse or large-scale settings.
- The method relies on numerous hyperparameters, yet the paper provides limited ablation studies or comparative experiments to assess their impact. This lack of analysis makes it difficult to determine whether the reported high accuracy stems from the intrinsic design of the FLEA block or from careful hyperparameter tuning, weakening the persuasiveness of the experimental results.
- Previous related works typically validated their methods on widely used architectures like ResNet, whereas this paper evaluates the approach solely on a self-designed network. This raises concerns about the practical applicability of the method and whether it can be effectively transferred or adapted to a broad range of existing architectures.

**Questions:**

- Could the authors elaborate on the potential practical applications of the proposed approach? Specifically, it would be valuable to clarify whether this method could be extended to scenarios where backpropagation incurs high memory or computational costs, such as the training of large-scale models or LLMs. If such deep and large models could be effectively trained in a layer-by-layer fashion, this would represent a significant contribution to the machine learning community.
In its current form, the manuscript appears relatively simple. For example, a straightforward baseline could involve injecting a projection at each layer or block to directly predict the label. While the label encoder/decoder design is novel, evaluating it solely on classification tasks may not fully demonstrate the method’s potential or effectiveness across broader applications. Further discussion and experiments would help clarify the generalizability and practical impact of the approach.

---

> ### Author Response · Authors · 2025-11-26
> **Response to Reviewer TDov (1/2)**
>
> We thank the reviewer for the detailed and constructive feedback. Below we address each point.
>
> ### **Response to weaknesses 1**
> Existing FF-based methods are almost exclusively evaluated on small and mid-scale datasets such as MNIST and CIFAR-10. Our experiments therefore follow the standard evaluation protocol and provide fair, meaningful comparisons. Within this setting, our method delivers substantial improvements over prior FF approaches, as demonstrated by our CIFAR-100 results, which already cover a much more complex task than previous baselines.
>
> Extending FF-based training to large-scale datasets is indeed an important future direction. A key challenge is the limited interaction between layers inherent to FF learning. We believe that our label-encoder mechanism may offer a promising pathway to enable such inter-layer communication and make FF methods more scalable.
>
> #### **Table: Ablation study**
>
> | **Configuration**                      | **# Param.** | **Train Acc.**           | **Val Acc.**             | **Test Acc.**            |
> |----------------------------------------|--------------|---------------------------|---------------------------|---------------------------|
> | DeeperForward (100 filters)            | 275K         | 50.20%                    | 49.10%                    | 49.11%                    |
> | Tiny (3-block; 64 filters) baseline    | 216K         | 50.26 ± 0.23%            | 54.58 ± 0.78%            | 55.13 ± 0.18%            |
> | No learnable threshold                 | 216K         | 50.19 ± 0.86%            | 54.62 ± 0.85%            | 55.09 ± 0.37%            |
> | Uniform LR & WD                        | 216K         | 52.14 ± 0.23%            | 55.90 ± 0.78%            | 56.34 ± 0.22%            |
> | Mean activation variant                | 216K         | 50.92 ± 0.14%            | 56.02 ± 0.45%            | 56.71 ± 0.26%            |
> | Fixed α = 1                            | 216K         | 50.41 ± 0.24%            | 55.59 ± 0.69%            | 55.78 ± 0.29%            |
> | Simple config                           | 216K         | 53.61 ± 0.20%            | 57.99 ± 0.58%            | 58.29 ± 0.32%            |
>
> *Table: Ablation study evaluating the effect of key hyperparameters and architectural choices in the proposed method. Mean and standard deviation are obtained over 10 runs with different random seeds.*
>
>
>
> ### **Response to weaknesses 2 and question 3**
>
> We appreciate the reviewer’s concern regarding hyperparameter sensitivity. As noted in the paper (line 431), our initial hyperparameter search was limited and was for CIFAR-10. In response, we performed a series of ablation studies to isolate the impact of key components, including the learnable threshold, hard negative mining, and per-layer learning-rate/weight-decay settings. The results are summarized in the table above.
>
> “Uniform LR & WD” uses a learning rate of 0.001 and weight decay of 0.0001 for all layers.
> “Simple config” removes the threshold, applies uniform LR/WD, uses mean activation, and selects a negative label by sampling from a multinomial distribution whose weights are given by the softmax of the class scores (excluding the true label and renormalizing). Notably, the Simple config already achieves a substantial boost over the baseline, demonstrating that the method’s effectiveness is robust and not reliant on finely tuned settings.
>
> The ablation results show that no single hyperparameter or component dominates performance. While individual changes produce small variations in training, validation, or test accuracy, the overall improvement comes from the combined design of the FLEA block rather than careful tuning of any particular hyperparameter.

---

> ### Author Response · Authors · 2025-11-26
> **Response to Reviewer TDov (2/2)**
>
> ### **Response to weaknesses 3 and question 1**
> We appreciate this insightful question. Indeed, most widely adopted architectures (e.g., ResNets, transformers) are fundamentally designed around backpropagation. For instance, residual connections rely on gradient flow through skip paths so that each block can learn a residual function relative to its input. Without a global error signal propagating backward, as in layer-wise training, residual additions reduce to summing largely uncoordinated feature maps, leading to degraded or unstable learning. A similar issue arises in architectures that depend heavily on cross-layer interactions, such as squeeze-and-excitation modules, and attention mechanisms. These components require coordinated optimization across multiple layers, which a purely layer-wise algorithm cannot provide.
>
> Therefore, we argue that architecture design and the learning algorithm must be considered jointly. While our method improves the accuracy of FF-based approaches, scaling such methods to large vision or language models would require new architectures explicitly designed for layer-wise optimization rather than direct adaptations of models built for backpropagation.
>
> The main motivation behind original Forward-Forward is the biological implausibility of backpropagation and also its huge memory demand during training which makes it especially inappropriate for training on devices with very limited resources. From this point of view, being able to train a CNN model  on devices with constrained RAM is a huge step forward.
>
> ### **Response to question 2**
>
> Our method is fundamentally an energy-based model (EBM). Introducing a classifier head with softmax and cross-entropy at each layer would instead convert the system into a probabilistic model, changing the underlying learning principle.
>
> As LeCun notes (*A Path Towards Autonomous Machine Intelligence*, p. 20):
>
> > “it is important to note that the definition of EBM does not make any reference to probabilistic modeling… the energy function is viewed as the fundamental object…”
>
> In EBMs:
>
> > “instead of trying to classify x’s to y’s, we would like to predict if a certain pair (x, y) fit together or not—find a y compatible with x.”
> > <https://atcold.github.io/NYU-DLSP20/en/week07/07-1/>
>
> This perspective motivates the label-encoder design and distinguishes our approach from simple layer-wise classification.
>
> We also acknowledge that extending the method to tasks beyond classification is an interesting and important direction for future work, but current literature on Forward-Forward is focusing on classification, and, as explained before, we already show a significant push in that regard with CIFAR-100.

---

### Official Review · Reviewer_Qc8q · 2025-10-31

**Soundness:** 2
**Presentation:** 2
**Contribution:** 2
**Rating:** 4
**Confidence:** 3

**Summary:**

Current neural networks are typically trained using backpropagation. To remove the backward pass computation, Forward-Forward training is previously studied. However, FF learning has a large accuracy drop compared to backpropagation. This paper proposes a feature-label embedding alignment (FLEA) to enhance the performance of FF training. FLEA introduces layer-wise discriminative learning, where each layer independently optimizes its parameters by aligning its feature embedding with the corresponding label embedding that maximizes class separability. Experiments demonstrate that FF with FLEA outperforms previous FF training methods.

**Strengths:**

1. A good background and explanation is provided for Forward-Forward learning.
2. The hard example learning is proposed.
3. The results show that the proposed FLEA outperforms previous FF method by large.

**Weaknesses:**

1. The explanation of Forward–Forward learning in section 3 provides a good background. However, the analysis is conventional, there is no tight connection between this explanation and the proposed method. Moreover, the analysis (specifically, the separability of equation 2) will not stand for other layers except the first layer.
2. The label encoder is basically the same with MLP classifer layer. In my understand, the proposed FLEA is a layer-wise classification learning method. From this point, the proposed method is similar with DeeperForward.
3. Some details are not clear, such as how to collect the layer-wise classification into the final classification, the gradient is only provided for features, but not for the weights to be learned.
4. The experiments use different network architectures, making the comparison not on the same basis and making it hard to evaluate the advantages of the proposed method. From the layer-wise classification viewpoint, concatenating features of previous layers (Appendix A.2) maybe the key for the performance improvement.

**Questions:**

1. What's the difference of the proposed method with a mlp classification on the representation encoder? What if CE loss is used instead of the positive-negative loss of line 361?
2. What's the difference of the proposed method with DeeperForward?
3. How the proposed method performs on simple feed-forward networks?
4. The FF method of Hinton 2022 also use hard negative labels, what are the differences?

---

> ### Author Response · Authors · 2025-11-26
> **Response to Reviewer Qc8q (1/2)**
>
> We thank the reviewer for the detailed and constructive feedback. Below we address each point.
>
> ### **Response to weakness 1**
> Our analysis aimed to show that Forward–Forward (FF) learning can be interpreted as aligning bottom-up feature extraction with a top-down inductive bias, and that our method generalizes this idea. In particular, Equations 5b (line 212), 6b (line 222), and 7b (line 233) mirror the alignment equations presented in lines 367–370. We will better clarify this connection in the revised version.
>
> We agree that our theoretical analysis does not extend beyond the first layer. As we hypothesize in lines 411–417, the supervisory signal (the top-down inductive bias) weakens rapidly with depth, which also explains why the original FF experiments were limited to very shallow networks (2–4 layers). This motivates our design choice: rather than relying on a single global signal, we inject a supervisory signal at every layer to compute a loss and update parameters.
>
> ### **Response to weakness 2 and questions 1 & 2**
> This is an insightful observation. Our method is an **energy-based model (EBM)**. Introducing a classifier head with softmax and cross-entropy at each layer would instead turn the system into a **probabilistic model**, altering the underlying learning principle.
>
> As noted by LeCun (*A Path Towards Autonomous Machine Intelligence*, p. 20):
>
> > “it is important to note that the definition of EBM does not make any reference to probabilistic modeling. Although many EBMs can easily be turned into probabilistic models, e.g. through a Gibbs distribution, this is not at all a necessity. Hence the energy function is viewed as the fundamental object and is not assumed to implicitly represent the unnormalized logarithm of a probability distribution.”
>
> In EBMs:
>
> > “instead of trying to classify x’s to y’s, we would like to predict if a certain pair of (x, y) fit together or not. Or in other words, find a y compatible with x.”
> > <https://atcold.github.io/NYU-DLSP20/en/week07/07-1/>
>
> Using classifier heads would also change the gradient flow. With classifier heads, gradients to the feature extractor must be obtained through the chain rule. In our formulation, the loss sits directly between the feature extractor and the label encoder, so both components receive gradients directly without passing through an intermediate classifier (lines 648–669).
>
> DeeperForward actually relies on softmax and cross-entropy, so as you mentioned it is a layer-wise classifier whose weights are one for specific feature-map groups and zero for the rest.
>
> #### **Table: classifier heads vs. label encoders**
>
> | **Configuration**               | **Train Acc.**       | **Val Acc.**         | **Test Acc.**        |
> |--------------------------------|-----------------------|------------------------|------------------------|
> | Local classifiers w/ CE loss   | 96.50 ± 0.09%         | 72.34 ± 0.75%          | 72.55 ± 0.25%          |
> | Encoders w/ BCE loss (ours)    | 93.75 ± 0.12%         | 72.46 ± 0.40%          | 72.61 ± 0.25%          |
> | Global classifier w/ CE (BP)   | 97.78 ± 0.05%         | 73.89 ± 0.66%          | 74.01 ± 0.30%          |
>
> We replaced our model with CE-based classifier heads at each layer. As shown above, CE classifiers achieve similar accuracy. Interestingly, our method achieves lower training accuracy while having comparable validation/test accuracy, suggesting better generalization. We also add the result for BP as the upper bound.
>
> ### **Response to weakness 3**
> Thank you for highlighting these ambiguities. Each layer produces its own prediction. During training, we select the final depth based on validation accuracy and model size, and discard deeper layers if they do not contribute improvements. Only the label encoder corresponding to the chosen final layer is retained.
>
> Gradients for convolutional kernels are computed from the feature-map gradients via standard cross-correlation. As for Hinton's FF, this reduces to a single operation for computing weight gradients from activations without needing to backpropagate through network blocks. We will revise the paper to clearly describe how predictions are selected and how gradients are calculated for the kernel weights.

---

> ### Author Response · Authors · 2025-11-26
> **Response to Reviewer Qc8q (2/2)**
>
> ### **Response to weakness 4**
> We performed an experiment using the publicly available DeeperForward implementation, using an architecture comparable to our CIFAR-100 setup (6-block model), with 600 filters per layer to match class multiplicity requirements (compared to 512 filters in our model). DeeperForward reached 63.90% test accuracy, compared to 72.61% with our method as reported above.
>
> #### **Table: Ours vs. DeeperForward**
>
> | **Configuration**                                 | **Train Acc.**       | **Val Acc.**         | **Test Acc.**        |
> |--------------------------------------------------|-----------------------|------------------------|------------------------|
> | Encoders w/ BCE loss (ours)                      | 93.75 ± 0.12%         | 72.46 ± 0.40%          | 72.61 ± 0.25%          |
> | Large (6-block; 600 filters) w/ DeeperForward     | 66.69%                | 64.58%                 | 63.90%                 |
> | Large (6-block; 500 filters) w/ DeeperForward     | 66.28%                | 63.58%                 | 63.40%                 |
>
> We acknowledge that architecture design strongly affects performance, especially in layer-wise training where no global loss exists. As Hinton notes (*The Forward-Forward Algorithm*, p. 5):
>
> > “what is learned in later layers cannot affect what is learned in earlier layers. This seems like a major weakness compared with backpropagation.”
>
> This prevents the effective use of advanced architectural components such as squeeze-and-excitation or even residual connections, since features from different layers cannot coordinate. Using label encoders to encourage inter-layer collaboration is a potential direction for future work.
>
> ### **Response to question 3**
> Our method applies to all types of layers. We trained a 5-layer MLP with 256 units per layer and obtained 99.26% test accuracy, compared to 99.39% using backpropagation on MNIST classification. Visualizations of the first-layer receptive fields for our model and for BP-trained models can be found at:
>
> - <https://ibb.co/Y41TM0fQ>
> - <https://ibb.co/5WVZBBjd>
>
> #### **Table: Ours vs. BP on MNIST (MLP)**
>
> | **Configuration**                     | **Train Acc.**        | **Val Acc.**          | **Test Acc.**         |
> |-------------------------------------|-------------------------|-------------------------|-------------------------|
> | 5-layer MLP (256 units), ours       | 99.48 ± 0.03%           | 99.24 ± 0.13%           | 99.26 ± 0.06%           |
> | 5-layer MLP (256 units), BP         | 99.66 ± 0.03%           | 99.35 ± 0.09%           | 99.39 ± 0.06%           |
>
> Note that in fully connected networks, activations contribute less to memory usage than in convolutional networks, where activations dominate due to weight sharing.
>
> ### **Response to question 4**
> Our experiments show that negative mining is crucial. As shown in Table below, sampling negative labels uniformly across incorrect classes reduces test accuracy by about 5%. Selecting the hardest negative deterministically causes later layers to collapse to near-random performance.
>
> #### **Table: Uniform vs. our negative mining**
>
> | **Configuration**               | **Train Acc.**       | **Val Acc.**         | **Test Acc.**        |
> |--------------------------------|-----------------------|------------------------|------------------------|
> | Encoders w/ BCE loss (ours)    | 93.75 ± 0.12%         | 72.46 ± 0.40%          | 72.61 ± 0.25%          |
> | Uniform negative mining         | 80.07 ± 0.30%         | 67.20 ± 0.75%          | 67.70 ± 0.25%          |
>
> Negative mining itself is not our contribution, but our results confirm that it is a key component of effective FF-style training, even for the proposed implementation.

---

### Meta-Review · Area_Chair_pBfp · 2026-01-07

**Summary:**

Reviewers agreed the paper targets an interesting direction (improving Forward-Forward / forward-only training for tiny, memory-efficient networks) and reports strong gains over prior FF baselines on MNIST/CIFAR. However, the decision is driven by consistent concerns that the technical novelty is limited (often resembling layer-wise classification/local heads under different framing), the theoretical justification is either weak or not clearly connected to deeper-layer behavior, and the experimental evidence is too narrow (mostly CIFAR-10/100 and MNIST, classification only, largely on a custom architecture) to support the broader claims about scalability, general applicability, and “backprop-free” impact.

**Reviewer Concerns:**

Addressed by rebuttal: authors clarified several implementation ambiguities (how layer-wise predictions are aggregated/selected, how convolutional weights receive updates from local gradients), provided additional comparisons (including to DeeperForward and a CE-based local-classifier variant), and added ablations/hyperparameter sensitivity evidence suggesting the gains are not from a single tuned trick.

Outstanding concerns: (1) novelty remains insufficiently differentiated from straightforward layer-wise supervised heads or prior FF variants, and the EBM framing does not translate into a clearly distinct capability; (2) theory remains unconvincing for deep networks (analysis largely does not extend beyond early layers, and the motivation for why the proposed alignment should reliably solve depth-related issues is not established); (3) experimental scope is still limited in datasets/tasks and does not demonstrate transfer to standard architectures (e.g., ResNet-like designs) or larger-scale settings; and (4) the paper’s positioning around being “backprop-free” and broadly practical is not fully supported without stronger baselines and broader validation.

**Reviewer Scores:**

Reviewer Qc8q: likely increases slightly (e.g., from borderline reject to weak accept / “would not mind accept”) given the added CE-head comparison, DeeperForward comparison, and clarified training/prediction details, but would still flag novelty and fairness-of-comparison/architecture-dependence concerns.

Reviewer TDov: likely remains near the original borderline-reject stance; rebuttal helps with ablations/clarity but does not resolve limited scope (no larger datasets/architectures) and the concern that improvements may be incremental.

Reviewer LHKM: likely remains near the original borderline-reject stance; rebuttal clarifies intent and adds some evidence, but the reviewer’s main issues (novelty and insufficient breadth of experiments/baselines) largely remain.

---

### Decision · Program_Chairs · 2026-01-26

Reject